# Simian Immunodeficiency Virus-Based Virus-like Particles Are an Efficient Tool to Induce Persistent Anti-SARS-CoV-2 Spike Neutralizing Antibodies and Specific T Cells in Mice

**DOI:** 10.3390/vaccines13030216

**Published:** 2025-02-21

**Authors:** Alessandra Gallinaro, Chiara Falce, Maria Franca Pirillo, Martina Borghi, Felicia Grasso, Andrea Canitano, Serena Cecchetti, Marco Baratella, Zuleika Michelini, Sabrina Mariotti, Maria Vincenza Chiantore, Iole Farina, Antonio Di Virgilio, Antonella Tinari, Gabriella Scarlatti, Donatella Negri, Andrea Cara

**Affiliations:** 1National Center for Global Health, Istituto Superiore di Sanità, 00161 Rome, Italy; alessandra.gallinaro@iss.it (A.G.); chiara.falce@guest.iss.it (C.F.); mariafranca.pirillo@iss.it (M.F.P.); andrea.canitano@iss.it (A.C.); zuleika.michelini@iss.it (Z.M.); 2Department of Infectious Diseases, Istituto Superiore di Sanità, 00161 Rome, Italy; martina.borghi@iss.it (M.B.); felicia.grasso@iss.it (F.G.); sabrina.mariotti@iss.it (S.M.); mariavincenza.chiantore@iss.it (M.V.C.); iole.farina@iss.it (I.F.); 3Confocal Microscopy Unit NMR, Confocal Microscopy Area Core Facilities, Istituto Superiore di Sanità, 00161 Rome, Italy; serena.cecchetti@iss.it; 4Viral Evolution and Transmission Unit, IRCCS Ospedale San Raffaele, 20132 Milan, Italy; baratella.marco@hsr.it (M.B.); scarlatti.gabriella@hsr.it (G.S.); 5Center for Animal Research and Welfare, Istituto Superiore di Sanità, 00161 Rome, Italy; antonio.divirgilio@iss.it; 6Center for Gender Medicine, Istituto Superiore di Sanità, 00161 Rome, Italy; antonella.tinari@iss.it

**Keywords:** virus-like particles, SARS-CoV-2, neutralizing antibodies, vaccine, spike, GAG, simian immunodeficiency virus

## Abstract

**Background/Objectives:** Virus-like particles (VLPs) represent an attractive platform for delivering vaccine formulations, combining a high biosafety profile with a potent immune-stimulatory ability. VLPs are non-infectious, non-replicating, self-assembling nanostructures that can be exploited to efficiently expose membrane-tethered glycoproteins such as the SARS-CoV-2 Spike (S) protein, the main target of approved preventive vaccines. Here, we describe the development and preclinical validation of Simian Immunodeficiency Virus (SIV)-based GFP-labeled VLPs displaying S from the B.1.617.2 (Delta) variant (VLP/S-Delta) for inducing persistent anti-SARS-CoV-2 neutralizing antibodies (nAbs) and S-specific T cell responses in mice. **Methods:** SIV-derived VLP/S-Delta were produced by co-transfecting a plasmid expressing SIVGag-GFP, required for VLP assembly and quantification by flow virometry, a plasmid encoding the Delta S protein deleted in the cytoplasmic tail (CT), to improve membrane binding, and a VSV.G-expressing plasmid, to enhance VLP uptake. Recovered VLPs were titrated by flow virometry and characterized in vitro by transmission electron microscopy (TEM) and confocal microscopy (CLSM). BALB/c mice were immunized intramuscularly with VLP/S-Delta following a prime–boost regimen, and humoral and cellular immune responses were assessed. **Results:** VLP/S-Delta were efficiently pseudotyped with CT-truncated S-Delta. After BALB/c priming, VLP/S-Delta elicited both specific anti-RBD IgGs and anti-Delta nAbs that significantly increased after the boost and were maintained over time. The prime–boost vaccination induced similar levels of cross-nAbs against the ancestral Wuhan-Hu-1 strain as well as cross-nAbs against Omicron BA.1, BA.2 and BA.4/5 VoCs, albeit at lower levels. Moreover, immunization with VLP/S-Delta induced S-specific IFNγ-producing T cells. **Conclusions:** These data suggest that SIV-based VLPs are an appropriate delivery system for the elicitation of efficient and sustained humoral and cellular immunity in mice, paving the way for further improvements in the immunogen design to enhance the quality and breadth of immune responses against different viral glycoproteins.

## 1. Introduction

Virus-like particles (VLPs) are nanoparticles of 20–200 nm in diameter, formed through the spontaneous polymerization of viral proteins in cell culture systems, mimicking the molecular and morphological features of authentic viruses, including size, shape and antigenic display features [1,2]. Depending on the characteristics of the parental virus, VLPs can be surrounded by a lipid bilayer (enveloped VLPs) or contained in rigid capsid structures (non-enveloped VLPs) [1]. VLPs can be quickly and efficiently produced on a large scale in various cell culture systems, including mammalian, plant, insect, yeast and bacteria cells, and have a high biosafety profile as they are non-infectious and do not have any replication potential within the target cells, due to the lack of genetic material. Therefore, they are an interesting tool for vaccinology, especially for the immunological advantages they offer [3,4]. Also, with their small size, VLPs can be easily captured by antigen-presenting cells (APCs), such as dendritic cells (DCs) and macrophages, and transported to the draining lymph nodes for the elicitation of cellular and humoral immune responses [5,6]. In this context, VLP-based vaccines have been licensed for clinical use against human papillomavirus [7], hepatitis B virus [8], hepatitis E virus [9] and Plasmodium Falciparum, and several additional VLP-based vaccine candidates are under evaluation [10,11].

The novel LNP-mRNA-based vaccines delivering Spike (S) glycoprotein have demonstrated great efficacy in reducing the severity and fatality of COVID-19. However, several studies have reported that the levels of vaccine-induced neutralizing antibodies (nAbs) declined over time [12] and that they were less effective against the subsequent variants of concern (VoCs), owing to increased transmissibility and a higher ability to evade the immune response compared to the Wuhan-Hu-1 ancestral strain [13,14,15,16]. In particular, the strong immune escape ability of the Omicron VoC and sub-variants required the administration of periodic and updated booster vaccinations [17], highlighting the necessity of continuous efforts to develop alternative vaccines. In this context, VLPs represent an attractive platform for a COVID-19 vaccine, since VLPs can be exploited to expose heterologous membrane-tethered pseudotyping proteins, like the SARS-CoV-2 homotrimeric envelope S glycoprotein. S protein represents the primary target of currently approved COVID-19 vaccines, being responsible for the virus’s entry into the host cells after binding with the angiotensin-converting enzyme 2 (ACE2) receptor on the surface of epithelial cells, especially in lungs and bronchial and nasal epithelia [18,19]. Therefore, the induction of functional and persistent nAbs directed to Spike remains the main goal of an effective vaccine against SARS-CoV-2.

Here, we report the development of a platform based on GFP-labeled VLPs derived from Simian Immunodeficiency Virus (SIV) for the implementation of a SARS-CoV-2 vaccine. The presence of the GFP reporter protein in the VLP scaffold allows the rapid and accurate quantification of VLPs by flow virometry. We selected SIV-based VLPs due to the ability of SIV-Gag protein to self-assemble into particles that can be efficiently pseudotyped with heterologous viral glycoproteins, as we and others previously demonstrated for parental lentiviral vectors (LVs) incorporating influenza hemagglutinin [20], the HIV envelope [21,22], SARS-CoV-2 S [23,24] and envelope glycoprotein G from vesicular stomatitis virus (VSV) [25]. In particular, the truncation of the S cytoplasmic tail (CT) resulted in the greater incorporation of the S protein on the vector particles, thus increasing the infectivity of S-pseudotyped LVs [23]. Furthermore, one immunization with SIV-based Integrase-Defective LVs (IDLVs) expressing S protein induced persistent nAbs against the Wuhan strain and cross-reactive nAbs against SARS-CoV-2 VoCs, including Alpha, Beta, Delta and Omicron BA.1, BA.2 and BA.4/5, and T cell immunity detectable until six months after vaccination [24]. In particular, here, we exploited VLPs for the delivery of the CT-truncated S protein from the B.1.617.2 (Delta) VoC of SARS-CoV-2 (VLP/S-Delta), which was the main circulating VoC in Italy at the time that this study was conceived. The Delta VoC appeared for the first time in India in October 2020 and was responsible for the second wave of deadly SARS-CoV-2 infections in 2021 in several countries worldwide, including Italy. At the time of its emergence, Delta was the most transmissible VoC [26,27,28], showing a 97% increase in transmissibility compared to the Wuhan strain, accelerated growth kinetics and higher levels of released virions [29]. Therefore, we hypothesized that a vaccine delivering VLPs pseudotyped with S from the Delta VoC in mice could result in the elicitation of Abs able to neutralize a broad spectrum of variants. To improve immunogenicity, VLPs were also pseudotyped with VSV.G since it has been shown that VLPs pseudotyped with VSV.G induced more efficient stimulation of immune responses, likely due to the VSV.G fusogenic activity into APCs that improves presentation on MHC molecules, relevant for the induction of specific immunity [30,31].

Here, we provide evidence that the S-Delta protein was efficiently exposed on the VLP surface and that the inclusion of GFP allowed us to easily quantify the amount of VLPs by flow virometry. The prime–boost immunization of mice with VLP/S-Delta elicited persistent anti-S Abs able to neutralize the homologous Delta VoC, the Wuhan-Hu-1 strain and the Omicron VoCs. An S-specific T cell response, including CD8+ T cells, was also induced.

## 2. Materials and Methods

### 2.1. Production of SIV-Based Virus-Like Particles (VLPs)

Simian Immunodeficiency Virus (SIV)-derived VLP/S-Delta were produced by the transient co-transfection of 293T Lenti-X cells (Clontech Laboratories Inc., Mountain View, CA, USA) as previously described [32]. Cells were grown in DMEM High glucose 4.5 g/L medium (Gibco, Life Technologies Italia, Monza, Italy) supplemented with 10% fetal bovine serum (FBS, Corning, Merk Life Science S.r.l., Milan, Italy) and 100 units/mL penicillin/streptomycin (P/S, Gibco). Briefly, 3.5 × 10^6^ cells were co-transfected with 8 µg pSIVGag-GFP [33] expressing the codon-optimized SIV Gag gene fused to the green fluorescent protein (GFP) from the pCDNA3.1 plasmid; 3 µg pSpike-INC3, which encodes the codon-optimized Spike sequence from the SARS-CoV-2 Delta variant (lineage B.1.617.2) with a 19-amino-acid deletion at the cytoplasmic tail (CT) and contains the mutations T19R, del157-158, L452R, T478K, D614G, P681R and D950N [24]; 3 µg of the phCMV-VSV.G_In_ [34] or phCMV-VSV.G_Co_ [35] plasmids expressing VSV.G from the Indiana (In.G) or Cocal (Co.G) serotypes, respectively, using the CalPhos^TM^ Mammalian Transfection Kit (Clontech Laboratories Inc.), following the manufacturer’s recommendations. Non-Spike-pseudotyped VLP/Mock were produced by co-transfecting 293T Lenti-X cells with 8 µg pSIVGag-GFP and 3 µg phCMV-VSV.G in the absence of the pSpike-INC3 plasmid. Forty-eight hours post-transfection, VLPs containing supernatants were collected, filtered with Millipore 0.45 μm filters (Millipore Corporation, Billerica, MA, USA) to remove any cellular debris and concentrated by ultracentrifugation on a 20% sucrose gradient at 65,000× *g*, for 2.3 h at 4 °C. VLPs were resuspended in phosphate-buffered saline (PBS, Gibco) and stored at −80 °C until use. A schematic flowchart of VLP production is depicted in Appendix A.

### 2.2. Flow Cytometry

First, 293T Lenti-X cells (3.5 × 10^6^) were plated in 100 mm dishes and transfected with the plasmid pSpike-INC3 (5 μg) using the JetPrime transfection kit (Polyplus Transfection, Illkirch, France). At 48 hrs from transfection, cells were stained with anti-S2 polyclonal antibody (Cat: 40590-T62, Sino Biological, Beijing, China; 1:3000) followed by donkey anti-rabbit PE (Cat: 406421, Biolegend, San Diego, CA, USA; 4 μg/mL) as a secondary Ab, or with anti-S1 human monoclonal antibody (1 μg/mL) COVA1-18 [36] followed by goat anti-human IgG secondary Alexa Fluor 488 (Cat: 109-545-006, Jackson ImmunoResearch, Ely, UK, 5 mg/mL) as a secondary Ab. The expression of Spike was measured with a Cytoflex S cytometer (Beckman Coulter Life Sciences, Brea, CA, USA). Results were analyzed with Kaluza 2.1 software (Beckman Coulter Life Sciences).

### 2.3. Confocal Laser Scanner Microscopy (CLSM)

The 293T Lenti-X cells (2.5 × 10^4^/well) were seeded in 24-well microplates onto 12 mm L-polylysine (Sigma-Aldrich, St. Louis, MO, USA)-treated cover glasses and transiently transfected with pSIVGag-GFP and pSpike-INC3 using the CalPhos™ Mammalian Transfection Kit (Clontech Laboratories Inc). Twenty-four hours after transfection, cells were washed and directly stained, prior to fixation, with anti-S2 polyclonal Ab (Cat: 40590-T62, Sino Biological, 1:50) followed by goat anti-rabbit IgG Alexa Fluor 595 (Cat: A-11072, Thermo Fisher Scientific, Waltham, MA, USA; 1:200) secondary Ab to detect membrane expression. The coverslips were rinsed, fixed with cold methanol and then mounted with Vectashield antifade mounting medium (Cat: H-1000-10, Vector Labs, Burlingame, CA, USA) on the microscope slides.

To detect the presence of Spike Delta protein on VLPs, 1.0 × 10^6^ concentrated VLP/S-Delta, produced with VSV-G In.G or Co.G, were seeded on clean glass coverslips treated with 10 μg/mL Polybrene (Millipore Corporation). After rinsing, coverslips were directly stained with anti-S2 polyclonal Ab (Cat: 40590-T62, Sino Biological, 1:50) and goat anti-rabbit IgG Alexa Fluor 595 (Cat: A-11072, Thermo Fisher Scientific; 1:200) secondary Ab prior to fixation with paraformaldehyde 3%. Confocal laser scanner microscopy (CLSM) images were acquired with the Zeiss LSM980 system (Zeiss, Oberkochen, Germany) and processed with Adobe Photoshop CS5 software programs (Adobe Systems, San Jose, CA, USA) and Zen Blue edition 3.3 (Zeiss, Oberkochen, Germany) as previously described [24].

### 2.4. Transmission Electron Microscopy (TEM) Analysis

VLP/S-Delta were produced after the transfection of 293T Lenti-X cells as described above. At 48 hrs post-transfection, 293T cells were incubated with anti-Spike COVA1-18 neutralizing mAb (1 μg/mL) [36]. This was followed by an anti-human IgG H and L (Goat, 10 nm Gold) which was used as a secondary Ab (Cat: ab39596, Abcam, Cambridge, UK; 1:50). Following staining, cells were fixed in cacodylate buffer 0.1 M, pH 7.2, with 2.5% glutaraldehyde and washed. Cells were post-fixed in the same glutaraldehyde/cacodylate buffer containing 1% OsO_4_. Fixed samples were embedded in an Agar 100 resin (Agar Scientific, Essex, UK), and ultrathin sections were recovered as described [21,24] and analyzed with a Philips 208S TEM at 100 kV.

### 2.5. VLP Quantification by Flow Virometry

The presence of GFP reporter protein fused to the carboxy-terminus of SIV-Gag protein allows for the visualization and quantification of VLPs by flow cytometry as previously described [37,38]. Briefly, concentrated GFP-labeled VLP stocks were serially diluted in PBS (Gibco) and acquired by the CytoFLEX LX flow cytometer (Beckman Coulter Life Sciences) using the Violet Side Scatter (V-SSC, 405 nm), necessary for the detection of particles smaller than 150 nm. A mixture of Megamix-Plus FSC and Megamix-Plus SSC (BioCytek, Marseille, France) reference beads was used to set the instrument, whereas PBS and non-fluorescent VLPs were used as negative controls to set the fluorescence threshold. Data were analyzed by using Kaluza 2.1 software (Beckman Coulter). The titers of VLPs were determined as the average of the results achieved by the acquisition of at least three dilutions in which the abort rate, indicating the events that the flow cytometer did not properly identify [39], was lower than 5%, as suggested by the manufacturer (www.beckman.it/resources/reading-material/posters/preparing-cytoflex-nanoscale-flow-cytometry).

### 2.6. Mouse Immunization Protocol

Six- to eight-week-old mice (BALB/c, female) were purchased from Charles River (Charles River, Calco, Como, Italy) and kept at the Istituto Superiore di Sanità (ISS, Rome, Italy) in the rodent facility under pathogen-free conditions. All procedures were authorized by the Italian Ministry of Health and reviewed by the Service for Animal Welfare at ISS (Authorization n. 731/2020-PR, 21 July 2020). Five mice/group were intramuscularly (i.m.) immunized with VLP/S-Delta pseudotyped with In.G (1.0 × 10^9^ VLP/mouse) and boosted 8 weeks later with the same VLP/S-Delta pseudotyped instead with the VSV Co.G non-cross-reactive serotype (1.0 × 10^9^ VLP/mouse). For negative controls, we injected five mice with VLP/Mock. Prior to immunization, we collected retro-orbital blood samples with glass Pasteur pipettes, repeating this after immunization at monthly intervals. Collected sera were stored at −80 °C until use. Sera were analyzed for the presence of anti-Spike Abs by neutralization assays and ELISA as described below. Six months after the first inoculum, mice were euthanized by CO_2_ inhalation using approved chambers. Spleens were harvested and processed for the analysis of cellular immune responses as described [24]. Briefly, single-cell suspensions of splenocytes were washed in complete RPMI medium (Gibco) containing 1 mM Na pyruvate (Gibco), 100 units/mL Pen/Strep (Gibco), 25 mM Hepes Buffer (Gibco), non-essential amino acids (Gibco) and 0.05 mM β-mercaptoethanol (Sigma-Aldrich) and supplemented with 10% FBS (Corning). Following low-speed centrifugation at 1500 rpm/4 °C/10 min, cells were suspended in complete medium, counted and stored in liquid nitrogen until further use.

### 2.7. Enzyme-Linked Immunosorbent Assay (ELISA)

Ninety-six-well plates (Greiner bio-one, Frickenhausen, Germany) were coated with recombinant RBD protein (rRBD) from SARS-CoV-2 (0.1 μg/well) as described [24]. After overnight incubation at 4 °C, plates were washed and incubated for 2 hrs at room temperature with PBS 1% BSA (Sigma-Aldrich) for blocking. After washing, serial 2× dilutions of mice sera were added to each well in duplicate and kept at room temperature for 2 hrs. After the incubation step, plates were washed, and to each well was added the biotin-conjugated goat anti-mouse IgG (Southern Biotech, Birmingham, AL, USA) followed by the horse radish peroxidase (HRP)-conjugated streptavidin (AnaSpec, Fremont, CA, USA). The reaction was incubated for 30 min in the dark with 3.3,5.5-tetramethylbenzidine substrate (TMB, SurModics BioFX, Edina, MN, USA) and stopped with 1 M H_2_SO_4_. Optical density at 450 nm wavelength (OD450) was measured by the Victor Nivo reader (PerkinElmer, Groningen, The Netherlands). Endpoint titers were determined as the reciprocal of the highest dilution with an absorbance value equal to at least threefold the values from naïve mice. The results are expressed as the log10 endpoint titer.

### 2.8. Production and Titration of Lentiviral Vectors Expressing Luciferase and Pseudotyped with Spike Variants

To produce SIV-based LVs expressing luciferase (LV-Luc) and pseudotyped with the Spike VoC, 3.5 × 10^6^ 293T Lenti-X cells were transiently transfected with plasmid pGAE-LucW expressing the luciferase, the packaging plasmid pAdSIV3+ [22] and each pseudotyping plasmid expressing Spike proteins, using the JetPrime transfection kit (Polyplus Transfection) following the manufacturer’s recommendations. pSpike-C3 [23], pSpike-INC3, pSpike-BA.1C3, pSpike-BA.2C3 and pSpike-BA.4/5C3 [24] expressing the cytoplasmic-truncated Spike from the Wuhan or Delta, Omicron BA.1, BA.2 and BA.4/5 VoCs, respectively, were used for LV-Luc pseudotyping. At 48 hrs post-transfections, culture supernatants with the LV-Luc pseudotypes were filtered using Millipore 0.45 μm (Millipore Corporation) and kept at −80 °C. The titers of S-pseudotyped preparations of LV-Luc were evaluated on Vero E6 cells as described [23,24]. Briefly, 2.0 × 10^4^ cells/well were plated in a 96-well white Viewplate (PerkinElmer) and transduced with 1:2 dilutions of LV-Luc pseudotypes. At 48 hrs from transduction, the expression of luciferase was measured with the Britelite plus Reporter Gene Assay System (PerkinElmer) using a Berthold Centro luminometer (Berthold Technologies, Bad Wildbad, Germany). Dilutions providing 2.0 × 10^5^ relative luminescence units (RLUs) were used in the neutralization assay.

### 2.9. Pseudovirus Neutralization Assay

Serum samples from immunized mice were heat-inactivated at 56 °C for 15 min and serially diluted starting from 1:80 and incubated with the LV-Luc/Spike pseudoviruses for 30 min at 37 °C in deep-well plates, 96-wells (Resnova, Roma, Italy), as previously described [23,24]. Pseudovirus- and cell-only controls were included. Samples were incubated with Vero E6 cells in 96-well Isoplates (PerkinElmer) (2.2 × 10^4^ cells/well). Luciferase activity was measured after 48 hrs with the Britelite plus Reporter Gene Assay System (PerkinElmer) using the Berthold Centro luminometer (Berthold Technologies). The serum dilution providing 50% inhibition of the infection (neutralization) compared to virus-only control wells is expressed as inhibitory dilution (ID) 50, calculated with a linear interpolation method [23,24].

### 2.10. IFNγ Enzyme-Linked Immunospot (ELISpot) Assay

The ELISpot assay was performed using the Mabtech ELISpot Mouse IFN-γ ELISpot flex (Mabtech, Stockholm, Sweden) following the manufacturer’s protocol. In brief, flat-bottom 96-well plates (Millipore) were coated O/N with anti-mouse IFNγ Ab. The next day, 2.5 × 10^5^ splenocytes/well from vaccinated mice were seeded in the plate and stimulated O/N with 1 μg/mL of a SARS-CoV-2 Wuhan Spike peptide pool of 15-mer sequences with an 11-amino-acid overlap covering the N-terminal S1 domain of the S protein (PepTivator^®^ SARS-CoV-2 Prot_S1, Miltenyi Biotec, Bergisch Gladbach, Germany), 5 μg/mL of H-2Dd-restricted SARS-CoV-2-Spike S-535-543 (KNKCVNFNF, S 9mer; ProImmune Ltd., Oxford, UK) [40,41] or H-2Kd-restricted GFP (HYLSTQSAL, GFP 9mer; ProImmune Ltd.) [42] epitopes, as specific stimulation, or using Concanavalin A (ConA; 5 µg/mL, Sigma-Aldrich) and medium as positive or negative controls, respectively. Spot-forming cells (SFCs) were counted with an ELISpot reader system (AID iSpot reader, AID GmbH, Strassberg, Germany). Results were indicated in the graphs as IFNγ-secreting cells (SFCs)/10^6^ cells. Samples were considered positive when at least 50 spots per 10^6^ cells were present and were 2-fold higher than the unstimulated cells.

### 2.11. Transduction of Human Monocyte-Derived Macrophages (HMDMs) and Vero E6 Cells

To construct the SIV-derived lentiviral transfer vector plasmid expressing mCherry fluorescent protein (pGAE-mCherry), the mCherry sequence was excised from pHTI-mCherry [33] using AgeI/EcoRI and cloned into the corresponding sites of pGAE-GFP [22,33]. In order to produce LV-mCherry/G or LV-mCherry/S, 3.5 × 10^6^ 293T Lenti-X cells were co-transfected with pGAE-mCherry, the SIV-based packaging plasmid pAdSIV3+ [22] and the plasmids expressing VSV.G [34] or the CT-truncated Spike glycoprotein from the Wuhan-Hu-1 strain (pSpike-C3) [23,24], using the JetPrime transfection kit (Polyplus Transfection), according to the manufacturer’s recommendations. The SIVGag-GFP-expressing plasmid [33] was included in the production of LV-mCherry to label the LVs with GFP and allow for flow virometry quantification and visualization by microscopy [21]. Supernatants containing the LV-mCherry were collected 48 hrs post-transfections, filtered with Millipore 0.45 μm filters (Millipore), concentrated by ultracentrifugation and stored at −80 °C until use. Recovered LV-mCherry/G or LV-mCherry/S were titrated by the RT activity assay and by flow virometry analysis, as described above, and used for the transduction of HMDMs.

Peripheral blood mononuclear cells (PBMCs) were isolated by Ficoll–Hypaque (Euroclone, Pero, MI, Italy) density gradient centrifugation from buffy coat preparations obtained from healthy donors. The isolated cells were resuspended in RPMI medium (Gibco) supplemented with 10% human serum (Millipore), 20% FBS (Euroclone), 10 mM HEPES (Euroclone), 1% P/S (Euroclone), 1% non-essential amino acids (NEAA, Euroclone) and 1% sodium pyruvate (Euroclone). PBMCs were cultured at a concentration of 1.0 × 10⁶ cells in 100 μL of supplemented RPMI for 5 days at 37 °C in an 18-well ibiTreat polymer coverslip μ-Slide (ibidi GmbH, Gräfelfing, Germany). Non-adherent cells were then removed by extensive washing with medium. HMDMs were subsequently cultured in RPMI (Gibco) complete medium and maintained in a controlled environment at 37 °C with 5% CO₂ in humidified incubators.

For the analysis of fluorescent proteins, 1.0 × 10^5^ Vero E6 cells (Cercopithecus aethiops-derived epithelial kidney, ATCC C1008) or 2.5 × 10^4^ HMDMs were cultured on a μ-Slide 18-well ibiTreat polymer coverslip (ibidi GmbH,) and transduced with 5.0 × 10^6^ particles of LV-mCherry/G or 2.0 × 10^7^ LV-mCherry/S. Forty-eight hours after transduction, cells were washed three times with PBS (Euroclone) and fixed with 4% PFA (Thermo Fisher Scientific) for 20 min. Hoechst 33,342 (Thermo Fisher Scientific) was added for 10 min to stain nuclei at a concentration of 1 μg/mL and then washed twice with PBS. Imaging was performed using a DeltaVision™ Ultra microscope (GE Healthcare, Chicago, IL, USA) equipped with either a 20× U Plan FI Dry Ph1 objective or a Plan Apo 60× (NA 1.42) oil immersion objective. Post-imaging analysis of the acquired images was carried out using SoftWorx and/or Fiji (ImageJ) version 2.8.0 software.

### 2.12. VSV.G Neutralization Assay

Titers of nAbs against Indiana or Cocal VSV.G were determined as previously described in Gallinaro et al. [21]. Briefly, LV-Luc/In.G and LV-Luc/Co.G were produced using 293T Lenti-X cells after transient transfection and were titered on Vero E6 cells. Sera from immunized mice were serially diluted starting from 1/100 and incubated for 30 min at 37 °C in deep-well plates, 96-well (Resnova) with pseudoviruses LV-Luc/In.G or LV-Luc/Co.G (2.0 × 10^5^ RLUs/well). Sera and LV-Luc mixtures were then added to Vero E6 cells plated on 96-well IsoPlate (PerkinElmer). The controls included LV-Luc only and cells only. After 48 hrs, Luciferase expression was measured as described above. Titers are expressed as ID50 calculated with a linear interpolation method [21,23,24], as described above.

### 2.13. Statistical Analysis

Data were analyzed with GraphPad Prism 9.4.1 (GraphPad Software Inc., San Diego CA, USA) using the one-way ANOVA test to compare three or more groups. *p* values < 0.05 were considered as indicative of statistical significance.

## 3. Results

### 3.1. Expression of S-Delta Glycoprotein and Psedudotyping on SIV-Derived VLPs

To verify that plasmid pSpike-INC3 is able to express the S protein, 293T Lenti-X cells were transfected with plasmid pSpike-INC3, and expression was evaluated by flow cytometry using anti-Spike antibodies. The 293T Lenti-X cells transfected with the plasmid expressing the S-Delta protein carrying the truncated cytoplasmic tail (CT) were stained with human anti-S1 nAb COVA1-18 or rabbit anti-S2 polyclonal Ab and analyzed by flow cytometry (Figure 1A). The results showed that CT-truncated S-Delta glycoprotein is efficiently expressed on the surface of transfected cells. Figure 1A represents the number of cells, expressed as percentage, which are positive for the Spike protein expressed by the pSpike-INC3 plasmid. The membrane localization of S-Delta protein was confirmed by confocal laser scanner microscopy (CLSM) observation of transfected cells (Figure 1B(a,b)). Importantly, when cells were co-transfected with both S-Delta- and SIV-Gag expressing plasmids, co-localization at the cell membrane level was evident (Figure 1B(e,f)), where SIV-Gag polyprotein is post-translationally targeted. The oligomerization of SIV-Gag polyproteins on the cell membrane (Figure 1B(c–f)) allows for the spontaneous budding of VLPs that can be enveloped by cell membranes containing the pseudotyping CT-truncated S-Delta proteins.

To verify S-Delta pseudotyping, VLP-producing cells were stained with anti-S1 COVA-18 mAb and visualized under transmission electron microscopy (TEM). VLP/S-Delta were produced with plasmids expressing vesicular stomatitis virus glycoprotein G from Indiana (In.G) or Cocal (Co.G) serotypes, as described in the Materials and Methods section. Released VLPs appeared as empty spherical particles presenting S-Delta glycoproteins on their surface (Figure 2A). These results were further validated by CLSM observation of VLP/S-Delta preparations after staining with anti-S2 polyclonal Ab followed by fluorescent anti-rabbit secondary Ab (Figure 2B). The GFP fused to the carboxy-terminus of SIV-Gag allows for the incorporation of GFP in the VLPs, which appear as green dots, while the signal emitted by the secondary Ab targeting the anti-Spike Ab is shown in red. The yellow dots in the merged images (Figure 2B, top panels) represent the co-localization of S-Delta with SIVGag-GFP, confirming that the VLPs were pseudotyped with the S-Delta protein.

### 3.2. Quantification of VLPs by Flow Virometry

In addition to allowing the visualization of VLPs by CLSM, the presence of the GFP reporter protein fused to SIV-Gag in the VLP scaffold enables the fluorescent VLP/S-Delta to be readily quantified by flow virometry [37,38]. Analysis was performed with a CytoFLEX LX cytometer, equipped with the 405 nm Violet side scatter (V-SCC) that enhances the sensitivity and resolution of the technique for the detection of small particles. Fluorescent beads of known diameter were used to standardize the settings and to identify the region between 100 nm and 200 nm, corresponding to the size of VLPs (Appendix A). PBS was used to set up the instrument background noise, due to the loss of scattered light during the acquisition of particles with much smaller sizes than cells (Figure 3, left panel). Scalar dilutions of the concentrated stocks of VLP/S-Delta and VLP/Mock resuspended in PBS were analyzed and counted to ensure that the percentage of missed events (abort rate), which resulted from the coincidence of two or more events detected as a single event by the instrument, was less than 5%. Only fluorescent events falling in the gate corresponding to 100 nm and 200 nm were counted. As shown in Figure 3, GFP-labeled SIV-derived VLPs (indicated as gated events in the rectangles) were efficiently produced with a high fluorescence intensity that allowed for clear separation from the background.

The concentrations of produced VLP/S-Delta were in the range of 10^10^ particles/mL (Table 1).

### 3.3. Prime–Boost Immunization with VLP/S-Delta Induced Anti-Spike Cross-nAbs in Mice

To assess the ability of SIV-derived VLPs to elicit specific and persistent anti-Spike immune responses in vivo, BALB/c mice (five per group) were immunized twice at 0 and 8 weeks intramuscularly (i.m.) with 10^9^ VLP/mouse of VLP/S-Delta or VLP/Mock. Titers of anti-RBD IgG in the sera of vaccinated mice were evaluated monthly by ELISA. The results showed that priming with VLP/S-Delta pseudotyped with VSV.G from the Indiana serotype elicited specific anti-RBD IgG Abs which were maintained over time and absent in the control mice. At 8 weeks after priming, all mice were successfully boosted with VLP/S-Delta pseudotyped with VSV.G from the Cocal serotype (Figure 4). The peak response was detectable at 4 weeks post-boost, and then, Ab levels slowly waned but remained stable at higher levels than pre-boost until 24 weeks after the priming, the experimental endpoint. As expected, no specific anti-Spike immune responses were detectable in mice vaccinated with VLP/Mock.

Sera from immunized mice were also tested for the presence of vaccine-induced nAbs, using a pseudovirus assay based on LVs expressing luciferase (LV-Luc) and pseudotyped with S protein derived from different VoCs [23,24]. All animals developed nAbs against Delta SARS-CoV-2, homologous to the vaccine sequence (Figure 5A). The kinetics of nAbs mimicked the trend of binding Abs, showing an approximately 20-fold increase two weeks after the boost (ID50 titer range at week 8: 199–978; ID50 titer range at week 12: 3424–10240). As expected, anti-S nAbs were absent in the sera of mice derived from the mock group. To evaluate the breadth of vaccine-elicited immune responses, sera from all vaccinated mice were also assayed for their cross-neutralization ability against different VoCs at 16 and 20 weeks after the priming. LV-Luc pseudotyped with S from Wuhan-Hu-1 and from Omicron (Om) BA.1, BA.2 and BA.4/5 VoCs were used in the neutralization assay. As shown in Figure 5B, VLP/S-Delta induced cross-nAb titers against all the tested VoCs. Anti-Wuhan nAb levels were similar to those of anti-Delta nAbs (*p* > 0.05). A significant reduction in nAb titers was evident for all the Omicron VoCs.

### 3.4. VLP/S-Delta Elicited Persistent T Cell Immunity

Twenty-four weeks after the first immunization, mice were sacrificed and the IFN-γ enzyme-linked immune absorbent spot (ELISpot) assay was performed in order to evaluate the efficacy of VLP/S-Delta vaccination in inducing a long-term Spike-specific cellular immune response (Figure 6). Fresh splenocytes were stimulated with a pool of overlapping peptides covering the N-terminal S1 domain of Wuhan S protein and the immunodominant H-2Dd-restricted S-9mer (KNKCVNFNF [40,41]) or the H-2Dd-restricted GFP-9mer (HYLSTQSAL [42]) peptides. As expected, a high GFP-specific response was present in mice immunized with either VLP/S-Delta or VLP/Mock, due to the presence of the GFP fused to SIV-Gag in the scaffold of the VLP. On the contrary, S-specific IFN-γ-producing T cells were detected only in mice vaccinated with S-pseudotyped VLPs, suggesting the ability of VLP/S-Delta to elicit effective and durable T cell immunity, including Spike-specific CD8+ T cells.

### 3.5. VSV.G-related Increase in Lentiviral Vector Uptake by Antigen-Presenting Cells

Previous work has shown that the incorporation of VSV.G into HIV-based vectors resulted in the more efficient stimulation of immune responses, likely due to VSV.G fusogenic activity into antigen-presenting cells, improving presentation on MHC molecules [30]. In this context, we evaluated whether VSV.G-pseudotyped SIV-based LVs were able to enter antigen-presenting cells (APCs) such as human monocyte-derived macrophages (HMDMs), in comparison to Spike-pseudotyped SIV-based LVs. To this aim, LVs expressing the mCherry fluorescent protein pseudotyped with either Spike (LV-mCherry/S) or VSV.G (LV-mCherry/G) were used to transduce both Vero E6 cells and HMDMs. The use of LVs was necessary to ensure the delivery of the mCherry reporter gene, the expression of which necessitates the entry and transcription of the LV genome in the target cells. To evaluate viral particle uptake in the absence of transgene expression, LVs were labeled with SIVGag-GFP, as described in the Materials and Methods. Transduced cells were analyzed under ultramicroscope to verify the expression of mCherry and GFP, the latter of which was included in the vector scaffold fused to SIV-Gag during vector production (Figure 7). As expected, both LV-mCherry/S and LV-mCherry/G were able to transduce ACE2+ Vero E6 cells (Figure 7c,d,g,h). Conversely, the expression of mCherry was detected in HMDMs only after transduction with LV-mCherry/G (Figure 7e,f) but not with LV-mCherry/S (Figure 7a,b), indicating that VSV.G was necessary for the transduction of macrophages. Of note, GFP was detected in HMDMs transduced with either LV, indicating that the particles were uptaken by the APCs. However, the HMDMs transduced with the LV carrying the VSV.G protein showed an increase in GFP+/mCherry+ cells. These results suggest that the presence of VSV.G on viral particles could enhance their uptake by APCs, promoting the elicitation of the immune response.

### 3.6. The VSV.G Exchange Strategy Allowed for an Efficient Boost

VSV glycoprotein G was included in the VLP/S-Delta design to increase immunogenicity by enhancing uptake by APCs, as described above. Although VSV.G induces high levels of autologous nAbs, which makes its use in a prime–boost regimen difficult, there is a limited induction of cross-nAbs to other VSV serotypes [43,44]. Indeed, in previous works, we successfully applied a VSV.G exchange strategy using two different VSV.G serotypes in prime–boost vaccinations with VSV.G-pseudotyped IDLV in non-human primates (NHPs) [21]. Based on these data and in order to limit the anti-vector immunity induced by the priming, VLP/S-Delta particles were pseudotyped with VSV.G from the Indiana serotype (In.G) for the prime and from the Cocal serotype (Co.G) for the second injection. The presence of anti-VSV.G nAbs in the sera of vaccinated mice was evaluated after each inoculum by a neutralization assay using LV-Luc pseudotyped with either In.G or Co.G [21]. All immunized mice developed autologous anti-In.G nAbs by 4 weeks after priming (titer range: 2861–10731), whereas no anti-Co.G nAbs were detected (Table 2).

Importantly, at the time of the boost, nAbs against In.G increased (titer range: 11,639–37,522), whereas anti-Co.G nAbs showed a very low titer in one mouse (ID50: 133) but remained under the threshold of detection in the other four animals. The boost with Co.G-pseudotyped VLP/S-Delta further increased the levels of anti-In.G and elicited anti-Co.G nAbs at 4 weeks after the boost. These data indicate that the prime with the In.G serotype induced limited cross-neutralization against the Co.G serotype, thus enabling successful boosting with VLP/S-Delta. Indeed, the anti-S nAbs increased after the boosting vaccination with VLP/S-Delta (Figure 5A), confirming that the use of VSV.G from two different serotypes is a successful strategy for repeated immunizations.

## 4. Discussion

In this study, we described the design, production and preclinical validation of SIV-based VLPs as a vaccine platform for the efficient delivery of S protein from the B.1.617.2 (Delta) variant of SARS-CoV-2 (VLP/S-Delta). We selected the S-Delta variant since it was the main circulating VoC in Italy at the time that this study began. However, this is a proof-of-concept study, showing the potentiality of this platform, and the S-Delta represents a prototype vaccine antigen. Further improvements in the design of the immunogen able to elicit nAbs against conserved Spike epitopes will be necessary to widen the potency and breadth of nAbs, making them able to cover a greater number of variants [45].

The production of SIV-derived VLPs was mediated only by the SIV-Gag protein, which is able to self-assemble into VLPs in the absence of other viral proteins. In contrast, the production of SARS-CoV-2-derived VLPs requires the simultaneous expression of nucleocapsid (N), envelope (E) and membrane (M) proteins [46]. Furthermore, SIV-based VLPs do not derive from a human pathogen and have low or absent pre-existing immunity, increasing the safety and immunogenicity profile of the platform.

Since SIV-based VLPs are enveloped by the cellular membrane, they can be functionally exploited for the delivery of membrane-anchored heterologous glycoproteins that, being produced on mammalian cells, will retain native post-translational modifications. In previous reports, we showed that SIV-based LV particles can be exploited for delivering membrane-anchored immunogens also after the pseudotyping of heterologous viral envelope glycoproteins on the vector’s particles in addition to VSV.G, including influenza, HIV-1 and SARS-CoV-2 [20,21,24]. In this report, SIV-based VLPs were pseudotyped with a CT-truncated S-Delta which reduces its recycling, resulting in the higher expression and density of S protein on the viral particles compared to full-length S [23], whereas VSV.G glycoprotein was included during production to facilitate VLP uptake by target cells, including APCs, such as macrophages. In contrast to other viral vector-derived platforms where repeated use is limited by pre-existing or induced vector immunity, SIV-based VLPs can be used in repeated immunizations by switching VSV.G serotypes in an exchange strategy to reduce anti-vector immunity associated with multiple injections.

The GFP was included in the design of the VLP platform by fusing the C-terminus of SIV-Gag protein in frame with the GFP sequence for practical reasons. The incorporation of GFP in the VLP scaffold allowed VLP quantification by flow virometry, a relatively recent application of flow cytometry for the detection and analysis of viruses [47,48]. Results showed that the VLP/S-Delta were highly fluorescent, facilitating discrimination from the non-fluorescent instrument background noise and abolishing the necessity to label the VLPs with fluorescent nanobeads or Abs [49]. GFP-labeled VLPs were also readily visualized by CLSM, thus facilitating the visualization of S-pseudotyped VLPs. In addition, the anti-GFP CD8+ T cell response can be easily evaluated in BALB/c mice, due to the presence of an immunodominant MHC-I-restricted epitope [33,42], allowing for the positive control of injection also in mock-vaccinated animals.

The results showed that the S-Delta protein was efficiently expressed in mammalian cells and that when expressed together, S-Delta and SIVGag-GFP co-localized at the membrane level, indicating that after budding from transfected cells, SIV-based VLPs acquired their external lipid envelopes and thereby incorporated the S-Delta protein. The S-Delta pseudotyping the VLPs was confirmed by both TEM observation of VLP/S-Delta-producing cells and CLSM analysis of the concentrated stocks of VLP/S-Delta.

The intramuscular priming of BALB/c mice with VLP/S-Delta effectively induced anti-RBD Abs and autologous nAbs in all immunized mice. Primed animals received a second administration of VLP/S-Delta that significantly boosted both the anti-RBD binding Abs and the anti-Delta nAb titers, which showed similar kinetics, with a slow decrease after the peak response, remaining at significantly higher levels than the pre-boost levels until 24 weeks, the endpoint of the study. Importantly, the prime–boost vaccination with VLP/S-Delta elicited high levels of cross-nAbs against the parental Wuhan-Hu-1 strain, which were comparable to those of anti-Delta nAbs, and anti-BA.1, -BA.2 and -BA.4/5 nAbs, although at levels that were significantly lower than the anti-Delta nAbs. This was expected since the genetic distance between the Delta and Omicron VoCs is higher compared to the distance between Delta and Wuhan strains [24,50].

The inclusion of the VSV.G glycoprotein in the S-Delta-pseudotyped VLPs may also have contributed to the VLP-induced immune responses. Indeed, VSV.G has a broad tropism that supports the transduction of a wide range of target cells, including APCs, such as DCs and macrophages [51]. In this study, we evaluated the contribution of VSV.G in the uptake of SIV-derived particles by evaluating the transduction potential of LVs expressing mCherry and pseudotyped with either S or VSV.G protein in HMDMs. The analysis of transduced cells by ultramicroscopy showed that the entry of lentiviral particles in HMDMs was improved by the presence of VSV-G. This also suggests that the inclusion of VSV.G in the design of VLP/S-Delta could have enhanced their uptake by APCs for the induction of a sustained immune response. Previous work has shown that the adherence of LVs pseudotyped with VSV.G to target cells leads to supplementary cycles of transduction over time in vitro and in vivo and that HIV-based VLPs pseudotyped with VSV.G showed higher immunogenicity in NHPs than VLPs that lacked VSV.G [30,52,53,54]. Furthermore, the presence of VSV.G on Gag particles allowed the processing of the antigens by both the MHC class I and class II pathways, eliciting both CD4+ and CD8+ T cell responses [55].

In order to avoid nAbs being directed to VSV.G of the Indiana serotype used for pseudotyping the priming VLP/S-Delta, boosting VLP/S-Delta were pseudotyped with VSV.G of the Cocal serotype. This VSV.G exchange strategy was adopted to allow for an efficient boost, as already described in NHPs vaccinated with SIV-derived vectors expressing HIV-Env [21]. Here, we confirmed the validity of the VSV.G exchange strategy and the efficiency of the vaccine platform, showing that boosting with VLP/S-Delta significantly increased both the anti-RBD binding Abs and nAb titers.

Immunization with VLP/S-Delta also elicited S-specific cellular responses, including CD8+ T cells, detected in all vaccinated mice at six months after the prime, indicating that our vaccine platform elicited efficient and persistent T cell immunity, in addition to humoral immune responses. Several studies have shown the important contribution of T cell immunity in the control of disease progression [56,57]. In particular, CD8+ T cells recognize and kill the virus-infected cells, providing local control of viral replication in the infected tissues and limiting the virus’s spread to distal organs, and therefore reducing the risk of multi-organ pathology and severe disease [58]. Furthermore, T cell escape from VoCs is an unlikely event, since the epitopes targeted by T cells are not restricted to the highly variable RBD and N-terminal domain and more than 80% is conserved among VoCs [59,60], suggesting a relevant role for T cells in protection from variants able to escape nAb recognition.

## 5. Conclusions

Here, we tested SIV-based VLPs as a vaccine platform for the delivery of S protein as a prototype antigen. Overall, the findings reported in this study provide evidence that SIV-based VLPs may represent an effective platform for the development of vaccines, able to induce specific and persistent cellular and humoral immune responses. The SIV-based VLP platform has the main advantages of being safer compared to genetic vaccines and more versatile than protein-based vaccines, especially considering the possibility of rapidly exchanging the vaccine antigen when viral variants occur.

## Figures and Tables

**Figure 1 vaccines-13-00216-f001:**
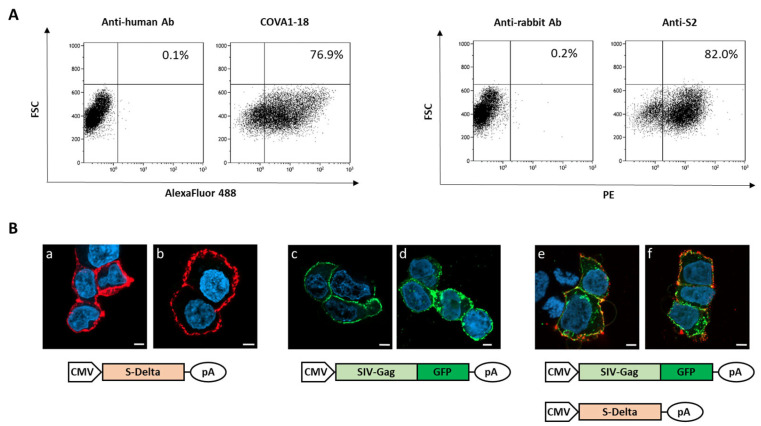
Analysis of S-Delta expression in transfected cells. (**A**) 293T Lenti-X cells were transfected with S-Delta-expressing plasmid (pSpike-INC3) and membrane stained with an anti-S1 neutralizing mAb (COVA1-18) or a commercial anti-S2 polyclonal antibody, and anti-human Alexa Fluor 488 or anti-rabbit PE secondary antibodies. The % of Spike-expressing cells is indicated in each plot. Representative results of 3 independent experiments are shown. (**B**) CLSM analysis of 293T Lenti-X cells transfected with S-Delta (panels **a**,**b**), pSIVGag-GFP (panels **c**,**d**) or both (panels **e**–**f**), and membrane stained with anti-S2 polyclonal antibody followed by anti-rabbit Alexa Fluor 594 secondary Ab (shown in red). Gag-GFP expression is shown in green. Blue color represents nuclei stained with DAPI. Scale bars are 5 μm. Two images are shown from one representative of 3 independent experiments.

**Figure 2 vaccines-13-00216-f002:**
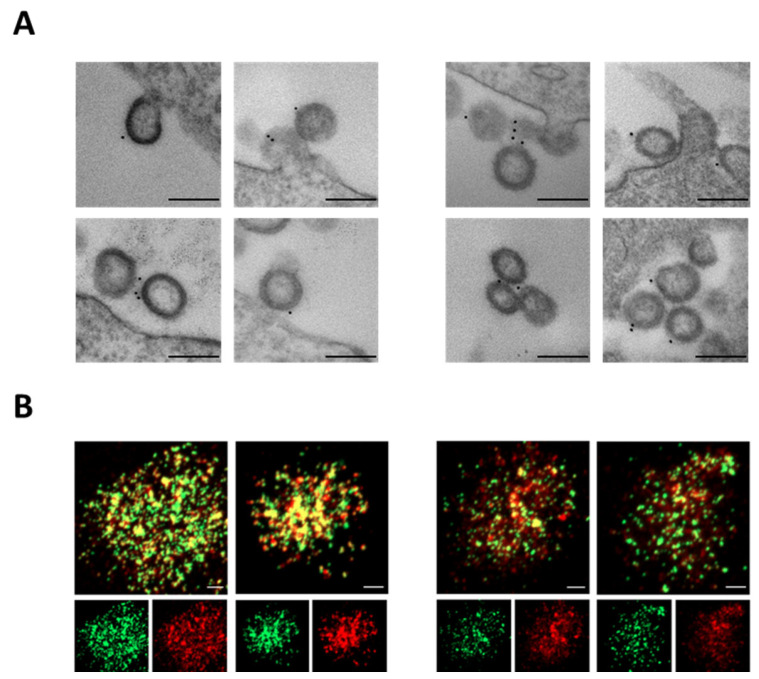
S-Delta protein is incorporated on VLP/S-Delta. (**A**) TEM observation of 293T Lenti-X cells transfected to produce VLP/S-Delta pseudotyped with In.G (left panels) or Co.G (right panels) after membrane staining with anti-S1 COVA1-18 mAb. Two representative images are shown for each sample. Bars: 0.2 μm. Results from one representative of 2 independent experiments are shown. (**B**) CLSM analysis of VLP/S-Delta produced with In.G (left panels) or Co.G (right panels) and stained with polyclonal anti-S2 Ab followed by secondary Ab anti-rabbit Alexa Fluor 594. Panels at the top depict the merged images. Yellow dots indicate the overlapping green (SIVGag-GFP) and red (S-Delta) signals. The panels at the bottom depict the single green or red fluorescent signal. Images from one representative of 2 independent experiments are shown. Bars: 5 μm.

**Figure 3 vaccines-13-00216-f003:**
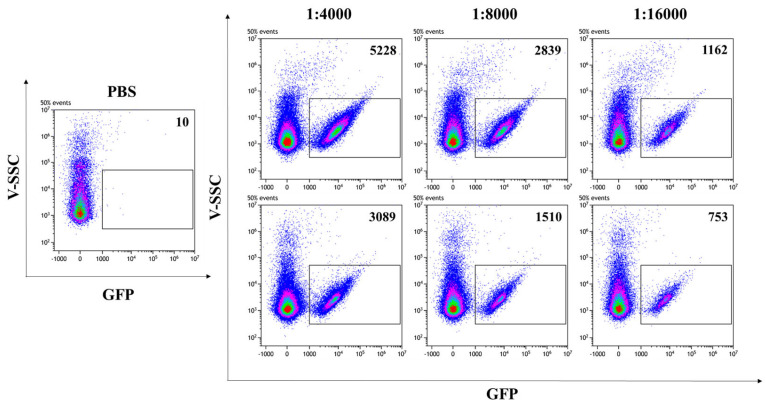
Flow virometry quantification of SIVGag-GFP VLPs. GFP-positive VLPs were serially diluted in PBS and analyzed with a CytoFLEX LX cytometer using the violet (405 nm) side scatter (V-SSC). Panels show the V-SSC vs. GFP (B525-FITC) density plots of PBS (left panel) and of three scalar dilutions of VLP/S-Delta pseudotyped with VSV.G from the Indiana (upper right panels) or Cocal serotypes (bottom right panels). The number of acquired GFP-positive events/μL is indicated in each plot. Shown is one representative of 3 independent experiments.

**Figure 4 vaccines-13-00216-f004:**
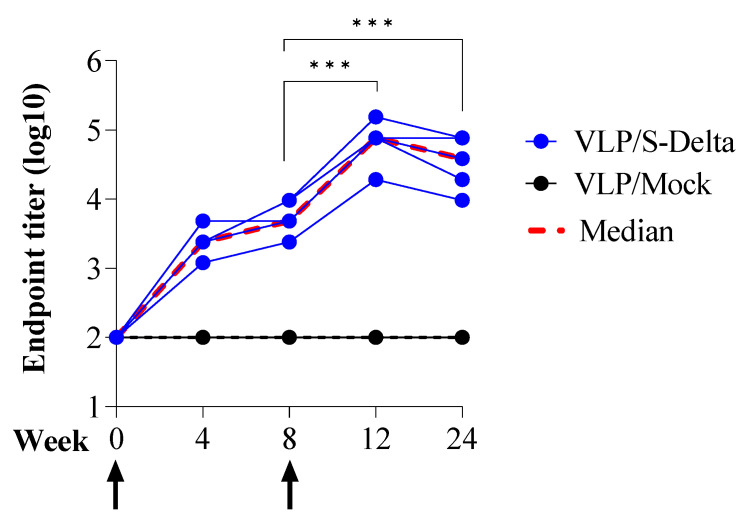
Prime–boost vaccination with VLP/S-Delta induced specific anti-Spike antibodies. Five BALB/c mice were intramuscularly immunized with VLP/S-Delta and boosted 8 weeks post-priming. Five mice immunized with non-pseudotyped VLPs (VLP/Mock) were used as negative controls. Blood was collected until 24 weeks after the first inoculum, the end of the study. Sera from vaccinated mice were analyzed for the presence of anti-RBD IgG Abs. Results are expressed as log10 endpoint titer. The black dotted line indicates the assay cut-off (minimum serum dilution tested—1:100). Arrows indicate the time of vaccination. *** *p* < 0.001; one-way ANOVA test.

**Figure 5 vaccines-13-00216-f005:**
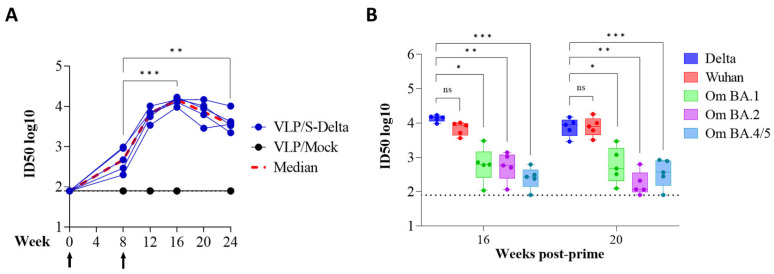
The kinetics of anti-Spike neutralizing antibodies (nAbs) in the immunized mice. (**A**) The anti-Spike nAbs (Delta VoC) were evaluated over a 24-week period at the specified time points. The results, expressed as log10 ID50, correspond to the dilution of sera with the RLUs reduced by 50% compared to control wells transduced with virus only. The dotted line (black) indicates the cut-off of the assay using a minimum serum dilution tested of 1:80. Arrows indicate the time of vaccination. (**B**) Neutralizing antibodies against the Wuhan ancestral strain and Omicron BA.1, BA.2 and BA.4/5 VoCs were evaluated at 16 and 20 weeks post-prime. Results are expressed as log10 ID50. Each dot represents a single mouse. The dotted line indicates the assay cut-off (minimum serum dilution tested— 1:80). *ns*: not significant; * *p* < 0.05; ** *p* < 0.01; *** *p* < 0.001; one-way ANOVA test.

**Figure 6 vaccines-13-00216-f006:**
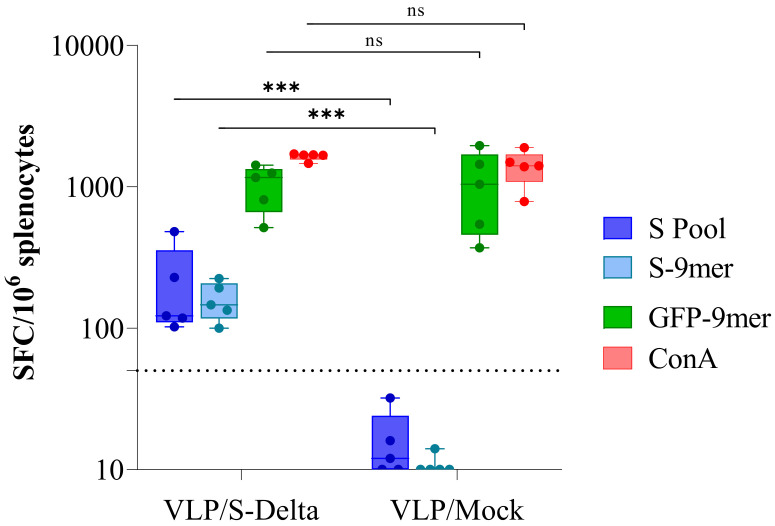
Analysis of Spike-specific T cell response. IFNγ ELISpot was performed 24 weeks after priming on splenocytes from mice immunized with either VLP/S-Delta or VLP/Mock. Cells were stimulated overnight with the Wuhan Spike peptide pool (S pool, blue boxes), the H-2Dd-restricted Spike (S-9mer, cyan boxes) or the H-2Kd-restricted GFP (GFP-9mer, green boxes) epitopes. Concanavalin A (ConA, red boxes) was used as a positive control. Data are expressed as mean specific spot-forming cells (SFCs) per million cells after background subtraction. Each dot represents a single mouse. The dotted line indicates the assay cut-off (<50 spots/10^6^ cells). *ns* not significant; *** *p* < 0.001; one-way ANOVA test.

**Figure 7 vaccines-13-00216-f007:**
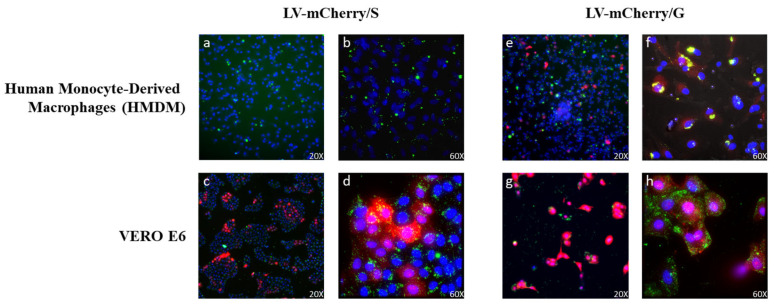
Lentiviral vector (LV) entry in APCs. HMDMs (top panels (**a**,**b**,**e**,**f**)) and VERO E6 cells (bottom panels (**c**,**d**,**g**,**h**)) were transduced with GFP-labeled LVs (green) expressing the mCherry protein (red) and pseudotyped with Spike (LV-mCherry/S, left panels) or VSV.G (LV-mCherry/G, right panels) and observed under an ultramicroscope. Shown are results from one representative of 2 independent experiments. Yellow represents the overlap of GFP and mCherry signals in double-positive cells. Nuclei are shown in blue.

**Table 1 vaccines-13-00216-t001:** Titers of concentrated stocks of VLP/S-Delta used for prime–boost immunization.

Sample	Dilution	Events	VLPs/µL	Average VLPs/µL	VLPs/mL
**VLP/S-Delta** **(In.G)**	1:1000	21,126 *	2.11 × 10^7^ *	1.87 × 10^7^	1.87 × 10^10^
	1:2000	9738 *	1.95 × 10^7^ *		
	1:4000	5228	2.09 × 10^7^		
	1:8000	2839	2.27 × 10^7^		
	1:16,000	1162	1.86 × 10^7^		
	1:32,000	388	1.24 × 10^7^		
**VLP/S-Delta (Co.G)**	1:1000	10,888 *	1.09 × 10^7^ *	1.21 × 10^7^	1.21 × 10^10^
	1:2000	5606 *	1.12 × 10^7^ *		
	1:4000	3089	1.24 × 10^7^		
	1:8000	1510	1.21 × 10^7^		
	1:16,000	753	1.20 × 10^7^		
	1:32,000	395	1.26 × 10^7^		

*** Values excluded from the titer calculation since the abort rate was >5%.

**Table 2 vaccines-13-00216-t002:** Neutralization activity against Indiana and Cocal VSV.G in sera from vaccinated mice.

Animal ID		IN1		IN2		IN3		IN4		IN5	
Vaccine Regimen	Week	In.G	Co.G	In.G	Co.G	In.G	Co.G	In.G	Co.G	In.G	Co.G
VLP/S-Delta (In.G)	0	<100	<100	<100	<100	<100	<100	<100	<100	<100	<100
	4	8950	<100	4619	<100	8045	<100	10,731	<100	2861	<100
VLP/S-Delta (Co.G)	8	17,740	<100	23,831	<100	37,552	<100	11,639	<100	22,466	133
	12	>62,500	953	22,047	<500	>62,500	13,517	>62,500	6417	>62,500	1045

## Data Availability

The raw data supporting the conclusions of this article will be made available by the authors on request.

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
