# Peer review of "Simian Immunodeficiency Virus-Based Virus-like Particles Are an Efficient Tool to Induce Persistent Anti-SARS-CoV-2 Spike Neutralizing Antibodies and Specific T Cells in Mice"

_vaccines, 2025, doi:10.3390/vaccines13030216_

Round 1

Reviewer 1 Report

Comments and Suggestions for Authors

The manuscript submitted by Alessandra Gallinaro descripted the function of Simian Immunodeficiency virus-based Virus-Like Particles, which could induce persistent anti-SARS-CoV-2 Spike neutralizing antibodies and specific T cells in mice immunization models. This data has certain reference significance and provides experimental basis for the development of a new SARS-CoV-2 vaccine.

However, there are also some limitations in the manuscript.

1.      As is known to all, with the high biosafety profile with a potent immune-stimulatory ability, VLPs represent an attractive platform for delivering vaccine formulations. The target gene is very important in the function of VLP. What are the characteristics of the target gene in this manuscript? What is the genetic variation of the target gene? Can this VLPs prevent and control clinical variant SARS-CoV-2?

2.      In 2.1 section, “µg of” should be replaced by “µg”. In other sections, if this situation occurs, it is recommended to revise it uniformly.

3.      Unit symbols are not standardized, such as “/ml” to be replaced with “/mL”, and “/µl” to be replaced by “/µL”.

4.      In 2.7 section, “H2SO4 1 M” should be revised to “1 M H2SO4”.

5.      In References section, some references lack page number, such as 1, 2, 6,7,11, et al.

6.      Is The DOI of reference necessary? In some reference, the doi is lack.

Author Response

REVIEWER 1

The manuscript submitted by Alessandra Gallinaro descripted the function of Simian Immunodeficiency virus-based Virus-Like Particles, which could induce persistent anti-SARS-CoV-2 Spike neutralizing antibodies and specific T cells in mice immunization models. This data has certain reference significance and provides experimental basis for the development of a new SARS-CoV-2 vaccine.

RESPONSE: We thank the reviewer for this comment.

However, there are also some limitations in the manuscript.

  1. As is known to all, with the high biosafety profile with a potent immune-stimulatory ability, VLPs represent an attractive platform for delivering vaccine formulations. The target gene is very important in the function of VLP. What are the characteristics of the target gene in this manuscript? What is the genetic variation of the target gene? Can this VLPs prevent and control clinical variant SARS-CoV-2?

RESPONSE: We thank the reviewer for this comment.

We selected the Spike protein from SARS-CoV-2 Delta variant of concern (VoC) because it was the main circulating VoC in Italy at the time this study was conceived (lines 96-103 in the manuscript).

We modified the Spike protein to include a 19 amino acid deletion at the cytoplasmic tail (CT) which we and others showed to improve the incorporation of the S protein on the vector particles (lines 90-92 in the manuscript). 

The Spike protein contains the following mutations compared to the parental Wuhan-Hu-1 Spike: T19R, del157-158, L452R, T478K, D614G, P681R, D950N (lines 126-127 in the manuscript).

It is difficult to deduce if this VLPs prevent and control clinical variant of SARS-CoV-2 since the VLPs were not tested in an efficacy trial. However, since the neutralization response against highly divergent Omicron VoC (BA.1, BA.2, BA.4/5 in the Figure 5B) was reduced compared to parental Delta Spike, we believe that further improvements in the design of the immunogen able to elicit protective nAbs against clinical variant of SARS-CoV-2 will be necessary to widen the potency and breadth of nAbs able to cover a greater number of variants (lines 539-541 in the manuscript).

  1. In 2.1 section, “µg of” should be replaced by “µg”. In other sections, if this situation occurs, it is recommended to revise it uniformly.

RESPONSE: We have modified the text accordingly.

  1. Unit symbols are not standardized, such as “/ml” to be replaced with “/mL”, and “/µl” to be replaced by “/µL”.

RESPONSE: We have modified the text accordingly.

  1. In 2.7 section, “H2SO4 1 M” should be revised to “1 M H2SO4”.

RESPONSE: We have modified the text accordingly.

  1. In References section, some references lack page number, such as 1, 2, 6,7,11, et al.

RESPONSE: The page number has been added in the references, when available, according to the citation style of each journal.

  1. Is The DOI of reference necessary? In some reference, the doi is lack.

RESPONSE: We have added the DOI in all the listed references.

Reviewer 2 Report

Comments and Suggestions for Authors

The COVID-19 pandemic has continued unabated, and the antigen has been changing rapidly, and there are concerns about the recent spread of avian influenza, making vaccine development a very important issue. In particular, there are concerns about IgG4 production in new mRNA vaccines, and newer technologies are needed. This paper is particularly important because it presents very hopeful results using VLPs. In particular, I commend the paper for providing reliable results using a variety of difficult techniques. VSV.G glycoprotein, it is also very promising that vaccines against various virus parts can be made. This is because the S protein has been mutated more and more, probably due to the vaccine, and is therefore difficult to use as a target.

I would like to point out the following points, and I would be happy if the authors would nudge me accordingly.

L116 2.1. Production of SIV-based Virus-like particles (VLPs)

Overall, it would be easier to understand if you could provide a simple illustration of how VLPs are constructed.

The plasmid pCDNA3-122 SIVGag-GFP does not seem to be well described in [34]. The structure of the mock plasmid should be easier to understand.

L190 The explanation of "abort rate" appears in L394, but since there is no explanation so far, I wonder what you are talking about. I would like to see an explanation here, or it should be given in the paragraph from L148.

L327, Fig. 5 The authors seem to be mixing non-parametric and parametric methods. In statistical testing, the answer depends on the method. So, it is not desirable for the analyst to arbitrarily choose a method, because it is less objective. Couldn't we have used ANOVA alone? Also, shouldn't the results of Fig.6 be tested?

L336 I would like an explanation of Figure 1A. What does this mean? Especially what is this percentage?

L340 Fig.1Bb I am concerned that the red signal is quite far from the cell surface. I would like some explanation for this. I imagine it is probably because the S protein is a reasonably long molecule as seen in the TEM of Fig. 2, and it is well expanded.

L388 Supplementary Figure 1A is at the end of the book, but it is a source of confusion, because it is confused with the one attached as the original image. It would be appreciated if you could write it as Supplementary Figure 1A "at the end of the article".

L398 Figure 3 GFP is divided into two plots, one with almost zero GFP and the other with some values, and the latter is surrounded by rectangles, but there is no explanation. I am sure this is not misleading, but it is difficult to understand, so I would like to see an explanation. What is the former one? Is it correct to understand that there are VLPs that do not take in GFP?

Fig.6 It is interesting to note that the GFP epitope is responsive to the GFP epitope here. I am glad that the result in Fig. 7 immediately following satisfies this curiosity; Fig. 7 should be titled "Spike VSV.G" instead of LV-mCherry/S, for example, to make it easier to understand.

L480 "APC such as HMDM" respectively should specify the abbreviation.

Author Response

REVIEWER 2

The COVID-19 pandemic has continued unabated, and the antigen has been changing rapidly, and there are concerns about the recent spread of avian influenza, making vaccine development a very important issue. In particular, there are concerns about IgG4 production in new mRNA vaccines, and newer technologies are needed. This paper is particularly important because it presents very hopeful results using VLPs. In particular, I commend the paper for providing reliable results using a variety of difficult techniques. VSV.G glycoprotein, it is also very promising that vaccines against various virus parts can be made. This is because the S protein has been mutated more and more, probably due to the vaccine, and is therefore difficult to use as a target.

RESPONSE: We thank the reviewer for this comment.

I would like to point out the following points, and I would be happy if the authors would nudge me accordingly.

QUESTION: L116 2.1. Production of SIV-based Virus-like particles (VLPs)

Overall, it would be easier to understand if you could provide a simple illustration of how VLPs are constructed.

RESPONSE: We thank the reviewer for this comment. In the new Supplementary Figure S1 we have included a depiction of how VLPs pseudotyped with Spike and mock VLPs, which are not pseudotyped with Spike, are constructed and their use.

QUESTION: The plasmid pCDNA3-SIVGag-GFP does not seem to be well described in [34]. The structure of the mock plasmid should be easier to understand.

RESPONSE: We apologize for the lack of clarity. The correct name of this plasmid is pSIVGag-GFP, as reported in the reference [34], now reference [33] in the revised manuscript. In this manuscript we named it by mistake “pCDNA3-SIVGag-GFP”, since the SIVGag-GFP fusion protein was cloned into the pCDNA3.1 plasmid (www.thermofisher.com/order/catalog/product/V79020). In the new version of the manuscript the plasmid is now named correctly pSIVGag-GFP. Construction of plasmid pSIVGag-GFP plasmid was described in the reference indicated and produces the SIVGag protein fused to the GFP protein from the pCDNA3.1 plasmid. This has been clarified in the text (lines 122-124). Mock-VLPs were produced as depicted in the new Supplementary Figure S1 by using only plasmids pSIVGagGFP and phCMV-VSV.G, producing the VLPs in the absence of pSpike-INC3 plasmid, which expresses the pseudotyping Spike protein. This is indicated in the text lines 133-135.

QUESTION: L190 The explanation of "abort rate" appears in L394, but since there is no explanation so far, I wonder what you are talking about. I would like to see an explanation here, or it should be given in the paragraph from L148.

RESPONSE: We apologize for the lack of clarity. The “abort rate” indicates the events that the flow cytometer does not properly identify. These events are removed by the cytometer from the number of counted particles. Since these events are eliminated from the data file, a high abort rate may contribute to misleading data. As a consequence, data files with an abort rate above 5% were not considered during our analysis. We included the explanation in the paragraph 2.5 with the new reference [40] and link to the manufactures.

QUESTION: L327, Fig. 5 The authors seem to be mixing non-parametric and parametric methods. In statistical testing, the answer depends on the method. So, it is not desirable for the analyst to arbitrarily choose a method, because it is less objective. Couldn't we have used ANOVA alone? Also, shouldn't the results of Fig.6 be tested?

RESPONSE: We apologize for the mistake. As pointed out by the reviewer, to be consistent, we can use the parametric ANOVA test instead of the Wilcoxon matched-pairs signed rank test. In the new version of the manuscript we included the new analyses using ANOVA and modified the text and figures 4 and 5a accordingly. Regarding Fig 6, we included the statistical analysis in the new Figure 6.

QUESTION: L336 I would like an explanation of Figure 1A. What does this mean? Especially what is this percentage?

RESPONSE: We apologize for the lack of clarity. The aim of the experiment shown in Figure 1 was to demonstrate that plasmid pSpike-INC3 is able to express the protein of interest (Delta Spike). For this reason, 293T Lenti-X cells were transfected with plasmid pSpike-INC3 and expression was evaluated by flow cytometry using an anti-Spike antibody. Figure 1A represents the number of cells, expressed as percentage, which are positive for the Spike protein expressed from the pSpike-INC3 plasmid after transfection 293T Lenti-X cells. This has been better clarified in the Results section, paragraph 3.1.

QUESTION: L340 Fig.1Bb I am concerned that the red signal is quite far from the cell surface. I would like some explanation for this. I imagine it is probably because the S protein is a reasonably long molecule as seen in the TEM of Fig. 2, and it is well expanded.

RESPONSE: As suggested by the reviewer the Spike protein is a reasonably long molecule. Figure 1B shows representative cells which are positive for the Spike protein (red signal) expressed from the pSpike-INC3 plasmid after transfection into 293T Lenti-X cells.

QUESTION: L388 Supplementary Figure 1A is at the end of the book, but it is a source of confusion, because it is confused with the one attached as the original image. It would be appreciated if you could write it as Supplementary Figure 1A "at the end of the article".

RESPONSE: We understand the reviewer concern. The editorial policy of the journal “Vaccines” requires to include the Supplementary Figures as separate files. For this reason we have removed the Supplementary Figures from the body of the manuscript. However, the final version of the manuscript published online will contain a weblink associated with the “Supplementary Figure S1” and with the “Supplementary Figure S2” that will link directly to each Figure after clicking on it.

QUESTION: L398 Figure 3 GFP is divided into two plots, one with almost zero GFP and the other with some values, and the latter is surrounded by rectangles, but there is no explanation. I am sure this is not misleading, but it is difficult to understand, so I would like to see an explanation. What is the former one? Is it correct to understand that there are VLPs that do not take in GFP?

RESPONSE: We apologize for the lack of clarity. We have added an additional panel on the left showing only PBS acquisition in the absence of GFP-labelled VLPs, representing the background acquisition signal/noise as visualized by the cytometer. The panels on the right show plots representing the cytometer acquisition of the GFP-labelled VLPs (indicated as gated events in rectangles), in which the noise is GFP negative. The numbers correspond to the VLP counts in each rectangle, which decreases with the dilution of each sample.

QUESTION: Fig.6 It is interesting to note that the GFP epitope is responsive to the GFP epitope here. I am glad that the result in Fig. 7 immediately following satisfies this curiosity; Fig. 7 should be titled "Spike VSV.G" instead of LV-mCherry/S, for example, to make it easier to understand.

RESPONSE: We apologize for the lack of clarity concerning Fig. 7. Panels on the left (a-d) represent macrophages (panels a and b) and VERO E6 cells (panels c and d) transduced with LV-mCherry pseudotyped with only Spike and without VSV.G (LV-mCherry/S). For this reason there is entry only in the VERO E6 cells, which are ACE2+.

Panels on the right (e-h) represent macrophages (panels e and f) and VERO E6 cells (panels g and h) transduced with LV-mCherry pseudotyped with only VSV.G and without Spike (LV-mCherry/G). For this reason there is entry in both macrophages and VERO E6 cells, since VSV.G allows the entry in both cell types.  The Fig. 7 has been modified by indicating the transduced cells.

QUESTION: L480 "APC such as HMDM" respectively should specify the abbreviation.

RESPONSE: A requested by the reviewer we have specified the abbreviation in line 475.